# Development and Validation of a Health Behaviour Scale: Exploratory Factor Analysis on Data from a Multicentre Study in Female Primary Care Patients

**DOI:** 10.3390/bs12100378

**Published:** 2022-10-02

**Authors:** Ewelina Chawłowska, Rafał Staszewski, Paulina Jóźwiak, Agnieszka Lipiak, Agnieszka Zawiejska

**Affiliations:** 1Department of Preventive Medicine, Poznan University of Medical Sciences, 60-781 Poznań, Poland; 2Department of Hypertension, Angiology and Internal Medicine, Poznan University of Medical Sciences, 61-848 Poznań, Poland; 3Department of Medical Simulation, Poznan University of Medical Sciences, 60-806 Poznań, Poland

**Keywords:** health behaviour, health promotion, scale development, questionnaire validation, factor analysis, primary care, women’s health, CVD patients, older women, 45+ women, health behaviour scale (HBS)

## Abstract

Health behaviours are the most important proximal determinants of health that can be either promoting or detrimental to the health of individuals. To assess and compare health behaviours in different socioeconomic groups within the population, a comprehensive, valid, reliable, and culturally appropriate measure is needed. This study aimed to develop a health behaviour questionnaire and validate it in a sample of female patients over 45 years of age with cardiovascular disease (CVD). The development procedure encompassed the following stages: literature search and item generation, content validity testing (focus group and expert evaluation), and field testing. A preliminary 38-item Health Behaviour Scale (HBS) was developed and tested in a group of 487 female primary care patients over 45 years of age. An exploratory factor analysis (EFA) yielded a four-factor structure. Factors jointly accounted for 47% of the variance observed. The results confirmed very good internal consistency of the questionnaire. The Cronbach’s alpha and McDonald’s omega coefficients for the entire scale were 0.82 and 0.84, respectively. The factor and item structure of the final 16-item HBS reflects the specificity of the studied sample. This measure can be a useful tool for primary care practitioners and public health researchers by helping them to develop interventions and strategies to reinforce health-promoting behaviours.

## 1. Introduction

Health behaviours (HB) are the most important proximal determinants of health. Pro-health behaviours are further affected by distal determinants, such as health policies, healthy settings, health promotion, and education interventions [1,2]. Healthy behaviours contribute not only to better health and well-being, but also to overall quality of life defined as “an individual’s perception of their position in life in the context of the culture and value systems in which they live and in relation to their goals, expectations, standards and concerns” [3].

Health behaviours are broadly defined as actions taken by individuals that affect their health or mortality. There are a number of health behaviour definitions [4,5], and for the purpose of this study, we adopted a definition by David S. Gochman (1997), who refers to HBs as demonstrated behavioural patterns, activities and habits that are related to health improvement, maintenance, or recovery [6]. The concept of health behaviour comprises both health promoting or health protective behaviours and unhealthy or health-risk behaviours. Health promoting behaviours include, for example, a healthy diet, physical activity, and adaptive coping strategies for stress management. In contrast, smoking, excessive alcohol consumption, and a sedentary lifestyle are regarded as unhealthy behaviours. Health behaviours are strongly socially patterned, with detrimental behaviours being less prevalent in higher socioeconomic status (SES) groups when compared to lower SES groups, where we can observe the clustering of risk behaviours [7,8,9,10]. This has serious health consequences, as both health protective and health risk behaviours are linked to all-cause mortality [11,12]. Modifiable behaviours, such as tobacco use, the harmful use of alcohol, an unhealthy diet, and physical inactivity increase the risk of noncommunicable diseases, which kill 41 million people each year, thus accounting for 71% of all deaths globally [13]. Cardiovascular disease (CVD) is a health problem where the influence of unhealthy behaviours has been found to be particularly strong: it is estimated to cause a 5-fold increase in CVD-attributed preterm mortality risk [14,15]. Still, there is evidence that effective interventions can be delivered through primary health care to strengthen early detection and timely, affordable treatment [16]. Primary care interventions which target nutritional behaviours appear to be the most effective in causing lifestyle changes [17].

Because the importance of health behaviours in disease prevention has long been recognized, many instruments for health behaviour measurement have been developed [18,19,20,21,22,23]. However, these measures are often focused only on particular groups of behaviours (e.g., diet, eating behaviours, physical activity, etc.), do not adequately measure the health behaviours of certain populations, or are outdated (e.g., do not take into account new risk factors and behaviours such as e-cigarettes, highly processed or fast foods, etc.).

Moreover, health behaviour instruments measure parameters that are usually embedded in the context of a particular population’s culture and language, which make both the adaptation and comparisons between countries difficult. Therefore, we decided to attempt to fill this gap and develop a comprehensive and up-to-date questionnaire—the Health Behaviour Scale (HBS)—comprising items related to preventive behaviours, the use of preventive services, health behaviours related to diet and physical activity, and unhealthy behaviours. Thus, a measure was proposed that could be used in the Polish population and whose results for various subpopulations could be compared and normalized. The aim of the present study was to develop a comprehensive questionnaire that could assess health behaviours and would have adequate psychometric properties.

## 2. Materials and Methods

### 2.1. Research Design

The presented quantitative instrumental study was based on a survey undertaken in 16 primary care centres which agreed to take part. The primary healthcare facilities were located in 4 provinces of Poland. Convenience sampling was used. The study group comprised 487 patients. The inclusion criteria were: (1) female sex, (2) age of over 45 years, (3) self-reported CVD, (4) being a patient of a primary healthcare facility, and (5) agreeing to participate. The study included a voluntary and anonymous paper-and-pencil survey assessing health behaviours. This questionnaire was distributed to patients by a healthcare professional (a primary care nurse or a family doctor). When in need of assistance in completing a survey form, patients were helped by professionals or accompanying family members. The respondents had an opportunity to ask questions while completing the questionnaire and to withdraw from the study at any point.

Informed consent was obtained from all subjects involved in the study. The Bioethics Committee of the Poznan University of Medical Sciences confirmed that the research did not constitute a scientific experiment and, as such, did not need ethical approval in accordance with the Polish law and Good Clinical Practice.

### 2.2. Development of the HBS

The stages of the process of development and content validation of the questionnaire are presented in Figure 1.

To generate the initial item pool for the questionnaire, we reviewed literature pertaining to:(1)the existing health behaviours scales to check what kind of items they contained and how robust the evidence supporting the items was [18,20,21,22,23,24,25,26,27,28,29,30,31];(2)the current definitions of health behaviours to see if they had any areas or domains not reflected in the existing scales [4,5,6];(3)the influence of health behaviours on such populations as women and persons with CVD to look for behaviours which need special attention [10,11,16,32,33,34,35,36].

The resulting item pool was screened for face validity by a focus group which consisted of lay people (10 office workers aged between 36–62 years of age). At this stage, four items which were too complex or difficult to answer were reworded. The content validation procedure was conducted on the basis of feedback provided by 5 experts with at least 10 years of professional experience in the field of public health, nursing, or preventive medicine. The experts assessed objectivity, accuracy, and comprehensiveness of the proposed measure, as well as relevance, accuracy, and clarity of each item. As a result, one item was removed, and one was added. Minor linguistic corrections and modifications to the remaining items were also made.

Furthermore, a list of thirty-eight items related to health behaviours was prepared. The items were formulated as closed-ended statements with a four-point scale response format. For some items a frequency scale was applied (always, often, sometimes, never or once a week, twice a month, once a year, depending on the statement) and for approximately the other half of the statements a positive-to-negative strength of agreement scale was used (yes, rather yes, rather no, no).

The resulting first version of the questionnaire consisted of 38 statements, which were divided into six domains. The domains focused on different groups of health behaviours: preventive behaviours related to the healthcare system—7 statements; individual preventive behaviours—7 statements; health behaviours related to eating—7 statements; health behaviours related to diet—7 statements; health behaviours related to physical activity—5 statements; and unhealthy behaviours—5 statements. Then, the first 38-item version of the HBS was used in the study. The HBS was finalized by decreasing the number of items to 16 as a result of content and construct validity assessments (for final HBS-16 with scoring please see Appendix A).

### 2.3. Statistical Analyses

An exploratory factor analysis (EFA) was used to attempt to identify internal attributes of the measure [37,38]. Bartlett’s test of Sphericity and the Kaiser-Meyer-Olkin’s (KMO) measure of sampling adequacy was first performed to assess if the items were significantly correlated and shared sufficient variance to justify factor extraction. The number of factors was determined based on the Kaiser criterion (eigenvalues greater than 1), examination of a scree plot, interpretation of factors and parallel analysis. The goodness-of-fit was assessed using the Tucker–Lewis Index (TLI) and the root-mean square error of approximation (RMSEA). TLI > 0.90, RMSEA < 0.06 indicated a good fit [39,40]. Internal consistency expressed as Cronbach’s alpha and McDonald’s omega coefficients was interpreted according to the principles specified by Robert DeVellis (2017) [41]. Correlations between factors were considered statistically significant at a 0.01 level. Statistical analyses were conducted using R version 4.2.1, R studio version 2022.2.3.492 [42], and SPSS 19.0 (SPSS Inc., Chicago, IL, USA).

## 3. Results

Factor analysis assessed the validity and factor structure of the measure and was performed in an adequately large sample of 487 female patients, with a very good participant-to-item ratio of 12:1 [43,44]. A preliminary 38-item HBS was developed and tested in a group of female patients with CVD over 45 years of age in primary care facilities. The mean age of respondents was 66.8 years (SD 11.5; Min. 45; Max. 89).

### 3.1. Exploratory Factor Analysis (EFA)

The construct validity of the measure was evaluated by performing an exploratory factor analysis. This EFA was used to examine the underlying factor structure of the new HBS and to identify true latent factors that explained common variance among HBS items. The EFA was performed using the maximum likelihood factor extraction method with the Oblimin rotation.

First, we checked whether the size of the sample was appropriate. Both Kaiser-Meier-Olkin (KMO) and Bartlett’s test of Sphericity returned results which confirmed that the sample size was suitable for further analysis (KMO = 0.87 and χ^2^ = 4949.04, *p* < 0.0001, respectively).

Next, we reduced the number of items in the HBS using the following criteria: items with correlation coefficients <0.4, items with an individual KMO <0.4, items with low factor loading <0.4, and cross-loadings were removed from further analysis.

Finally, we obtained a set of 16 items. The overall KMO value was 0.87. Factorial analysis with principal axis factoring, Oblimin rotation, and Horn’s parallel analysis identified four factors explaining 47% of the variance (see Table 1).

An initial examination of eigenvalues suggested a 4-factor solution as seen on the scree plot (Figure 2). All four extracted factors had eigenvalues above 1 (see Table 1).

Four factors were identified covering the range of HBs (see Figure 3). The first factor: Diet and Mental Health (F1) explained 22% of variance (see Table 1). It was comprised of 8 items including diet, nutritional/eating behaviours, relaxation, sleep, and stress with factor loadings ranging from 0.49 to 0.85. The second factor, Individual Healthy Behaviours (F2), which explained 10% of variance, was comprised of 4 items related to looking for information on diet, checking the labels of food products, checking the body for physical lesions and abnormalities, and performing breast self-examinations. Items in F2 have factor loadings ranging from 0.42 to 0.64. The third factor, Preventive Behaviours (F3), which explained 8% of variance, comprised 2 items on performing blood sugar and cholesterol tests. F3 items have factor loading ranging from 0.67 to 0.84. Finally, the fourth factor, Physical Activity (F4), which explained 7% of variance, had 2 items with factor loadings 0.72 and was related to two behaviours: leading an active lifestyle and using daily activities as an opportunity for physical activity.

Model fit was acceptable to good: RMSEA = 0.049, 95% CI [0.037–0.062], TLI = 0.947.

### 3.2. Reliability Analysis

Next, we computed Cronbach’s alpha and McDonald’s omega coefficients to estimate the internal consistency for the whole questionnaire and for each factor separately (see Table 2). Total Cronbach’s alpha for the 16-item questionnaire was 0.824 (95% CI: 0.818–0.830), while values for particular domains ranged from 0.68 for F1 (Physical Activity) to 0.76 for F3 (Preventive Behaviours). MD-omega total for the instrument was 0.84 and the values for particular factors, as presented in Table 2, ranged from 0.69 to 0.87. Thus, the internal consistency as measured with Cronbach’s alpha indices could be interpreted as very good for the whole scale and either minimally acceptable or respectable for particular factors. According to McDonald’s omega values, the internal consistency of the whole scale ranks as very good, and of particular domains—as minimally acceptable, respectable, or very good.

### 3.3. Construct Validity and Descriptive Data

Next, we assessed the factor correlation matrix of the final EFA to check the construct validity. The results are presented in Table 2. The internal correlations between the four factors, according to Pearson’s correlation coefficient, were all significant (*p* < 0.01), both positive and negative, with *r* ranging from −0.03 to 0.57. The largest positive correlations were between Factor 1 (Diet and Mental Health) and Factor 3 (Preventive Behaviours), and between Factor 2 (Individual Healthy Behaviours) and Factor 4 (Physical Activity); 0.57 and 0.45, respectively. The only negative correlation between factors was between F3 and F4 and was minor (−0.03). There were no correlations among factors exceeding 0.7; thus, we can conclude that the factors yielded by EFA have adequate discriminant validity.

Additionally, a descriptive analysis was carried out to determine what the respondents’ most frequently declared health behaviours were. Table 2 presents the factors’ arithmetic means and standard deviations for the total sample. The maximum number of points for subscale F1 was 24 (8 statements and for each statement the respondent could receive 3 points), for Factor 2 the maximum score was 12 points, and for F3 and F4–6 points. Relatively higher mean values of reported health behaviours were found for F1, and the lowest level of health behaviours was reported for F2.

## 4. Discussion

Numerous health behaviour questionnaires have been developed and validated to cater to the needs of various populations, age groups, and cultures. At times, these scales have assessed all kinds of health behaviours while some have focused only on specific aspects; some are up-to-date, while other ones are rather obsolete; and they can be universal or may serve one particular purpose [18,23,25,26,28,29,30,31]. We chose to develop a questionnaire that would measure the various types of health behaviours that could be applicable to the general population as well as to chronically ill subjects. Initially, the HBS comprised statements grouped in six domains: describing individual healthy behaviours, preventive behaviours, eating behaviours and diet, physical activity, and unhealthy behaviours. Knowing the significant differences between the health behaviours of men and women [32,33,34,45], we first attempted to validate this measure among female participants. What we wanted to achieve was to combine 6 dimensions of health behaviours and find out which items and factors would remain after conducting statistical analyses on responses from 487 women aged 45 and above. The initial pool contained 38 items, but, after the elimination of cross loadings and low factor loadings, 16 total items remained.

In addition to Cronbach’s alpha, we also assessed consistency using McDonald’s omega coefficient. It should be noted, however, that the use of MD-omega, though sound from a methodological point of view [41], is not frequent in validations of health behaviour scales. The few instrument development studies in which MD-omega was computed either do not report MD-omega total or values for all factors [46,47]. The overall internal consistency of the new HBS-16 as measured by Cronbach’s alpha, was very good and similar to other health behaviour measures [23,27,29]. Moreover, the coefficients for four factors of HBS-16 were much higher (0.68–0.76) than those of the HBI measure most frequently used in Poland (0.60–0.65) [23]. The new four-factor structure explained 47% of variance, which, in comparison to other validated health behaviour questionnaires, is a good result. For example, four factors of HBI and ten factors of Health-Related Behavior Scale explained 38.58% and 39.3% of variance, respectively [23,29].

The EFA revealed four latent domains: Diet and Mental Health, Individual Health Behaviours, Preventive Behaviours, and Physical Activity.

The first domain of the HBS-16 comprises of statements on coping with stress, finding time to relax, and sleep. High factor loadings of these items may show the general tendency to perceive mental health as an inherent part of general health, especially in female subjects who are particularly burdened by mental health problems [35].

Some items, namely “I perform a Pap smear test” received insufficient item loadings, whereas other items, such as “I perform a breast self-examination”, were successfully included in the final version of the measure. This is in accordance with other studies showing that breast self-examinations and mammography are more frequently performed among postmenopausal women than Pap smear tests [48] and that the frequency of gynaecological examinations decreases with age [34,49].

The final version of the HBS-16 included two items related to physical activity (“I lead an active lifestyle” and “I use daily activities as an opportunity for physical activity”). The two behaviours might be perceived differently: the former would denote all kinds of activity including sports, work life, socialising, travelling, etc., whereas the latter could pertain to physical activity per se, even though the activity might be undertaken “by the way” rather than on purpose. Another item referring to physical activity: “I engage in recreational physical activities [e.g., running, cycling, swimming, aerobics etc.]” was removed due to a low loading factor. This finding is supported by Shaw et al., 2010 and Holahan et al., 2020, who reported that age was negatively associated with leisure-time physical activity among middle-aged and older women [50,51]. Also, it can be predicted that our respondents, who suffer from CVD, may perceive engaging in physical activity as risky to their condition [52].

Also, the demographic characteristics of the sample resulted in the removal of items such as “I limit the number of cigarettes smoked” and “I smoke e-cigarettes”, as women 45+ are less likely to use e-cigarettes [53] or smoke than younger women [54] and men [55]. Similarly, all items related to alcohol consumption did not receive sufficient item loadings and were not included in the final version of the HBS-16, which is supported by the finding that women drink 3 times less alcohol than men and that the percentage of current drinkers decreases with increasing age [56]. For instance, it may be less socially acceptable for women, especially older women, to use tobacco or drink alcohol [57]. Also, as our respondents were CVD patients, it is likely that, by being health literate, they gave up smoking and drinking [58] or underreported these behaviours [16].

The relatively poorer, but still acceptable, internal consistencies of the F2 and F4 domains could be attributed to either the smaller number of items in those domains or, possibly, the greater inherent heterogeneity of health behaviours. It is possible that the respondents tried to respond to some of these statements in a socially acceptable manner, which affected the correlations with the rest of the items in the domains. This was also noticed in a study conducted by George et al., 2006 [59].

The HBS-16 is a self-reported measure appropriate for middle-aged and older women with concomitant CVD. It is very important that this population is thoroughly investigated and targeted with tailored public health interventions as the available evidence suggests that these women have unmet health and clinical needs [36].

Finally, it is also necessary to test the initial 38-item HBS in other populations such as with men or other age groups as the evidence shows those groups may be significantly different in terms of health behaviours. Therefore, we suspect that different latent factors could emerge in factor analyses and that the final measure would be composed of different items.

The HBS-16 items specifically reflect the most important health behaviours of the studied group, namely chronically ill women aged 45+. The scale is short and easy to navigate, and its statements are simple and easy to understand. We also hope that the future use of HBS-16 may enable the assessment of previously latent elements in relation to variables of socioeconomic status. This, in turn, can help to identify the obstacles in engaging women aged 45+ in healthy behaviours.

Additionally, the scale can prompt respondents to consider various ways of improving their health by drawing their attention to diverse healthy behaviours reflected in the scale items. The items may not only highlight what is lacking but also draw attention to potential pro-health behaviours of the respondents.

## 5. Limitations

Although a 16-item version of the questionnaire allows for rapid health behaviour assessment, it also has a limited scope as some aspects were not included in the final measure due to sample characteristics (female CVD patients aged 45+). It resulted in the removal of items such as: “I limit the number of cigarettes smoked”, “I smoke e-cigarettes”, “I go to the dentist for preventive check-ups”, and “I perform a Pap smear test”.

All self-reported measures like the one presented here may have limited validity due to recall bias and social desirability bias.

Generalisability of our results is limited for three reasons: the use of convenience sampling, the homogeneity of the sample (consisting of female CVD patients aged 45+), and the limited amount of the collected sociodemographic data (sex and age only).

The limitation of the presented study was its cross-sectional design. It is necessary to run a longitudinal study to find out whether higher HBS-16 scores are associated with better health, higher quality of life, or more effective prevention of non-communicable diseases.

Finally, the study presents preliminary validation of the scale and additional analyses are needed, e.g., confirmatory factor analysis, scales triangulation.

## 6. Conclusions

The proposed HBS-16 is a self-administered health behaviour assessment tool whose psychometric properties were validated in an adequately large sample of 487 women over 45 years of age, all with CVD. This scale incorporates 16 items where the respondent marks the frequency of health behaviours and her agreement with health-related statements. This measure is divided into four subscales: Diet and Mental Health, Individual Healthy Behaviours, Unhealthy Behaviours, and Physical Activity. The HBS-16 has good validity and reliability and is potentially applicable to both clinical practice and research.

In order to successfully use the HBS in populations other than the one in which it was validated, it may be necessary to develop and validate separate versions of the HBS that are gender-specific and age-specific because the evidence implies that a “one size fits all” approach may not work in health behaviour research.

## Figures and Tables

**Figure 1 behavsci-12-00378-f001:**
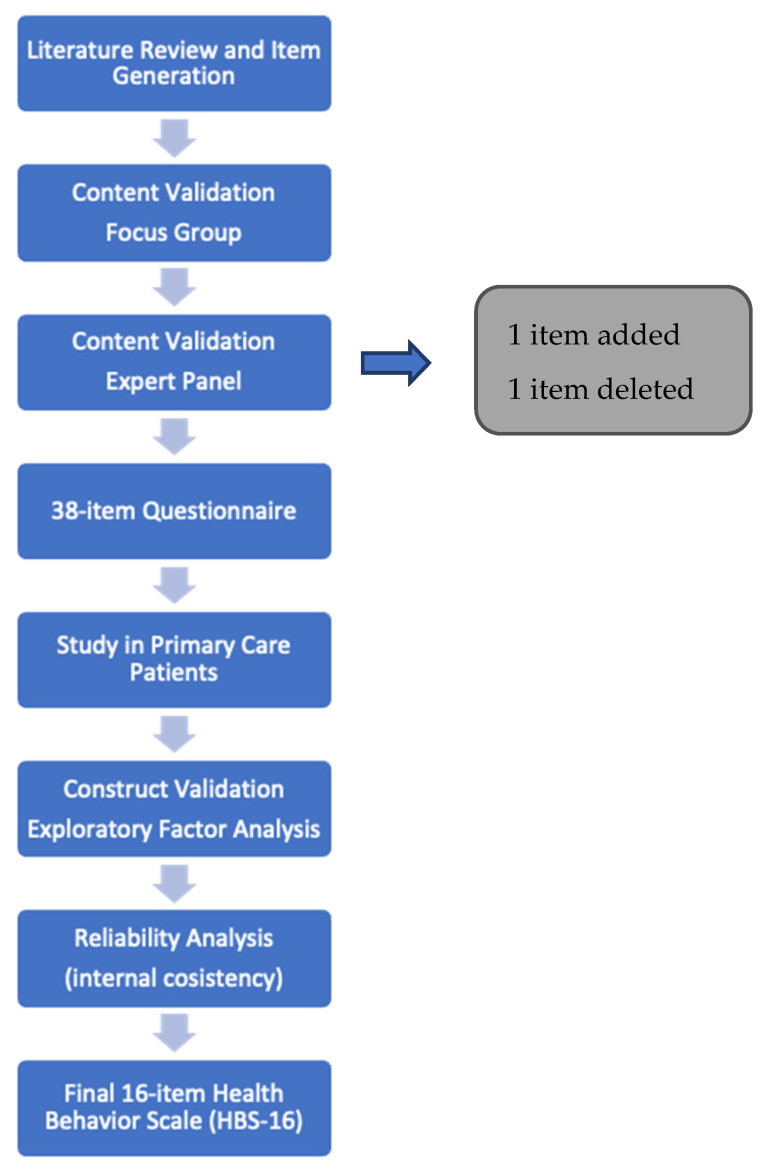
Health Behaviour Scale development and testing.

**Figure 2 behavsci-12-00378-f002:**
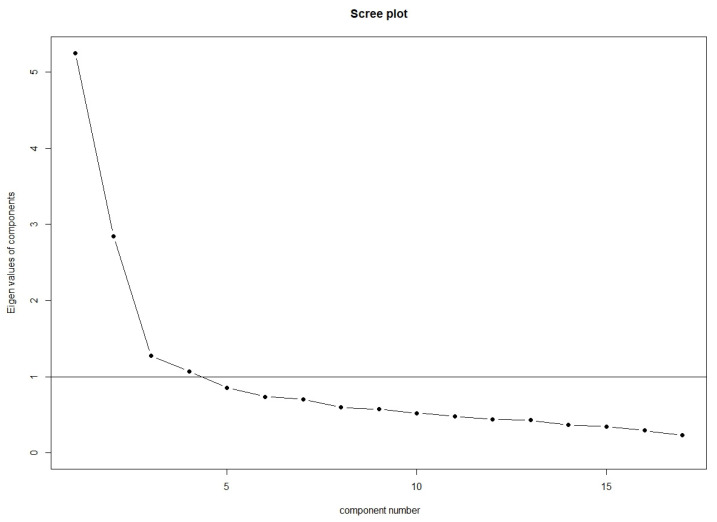
Scree plot of factors. Kaiser criterion is shown by the black horizontal line.

**Figure 3 behavsci-12-00378-f003:**
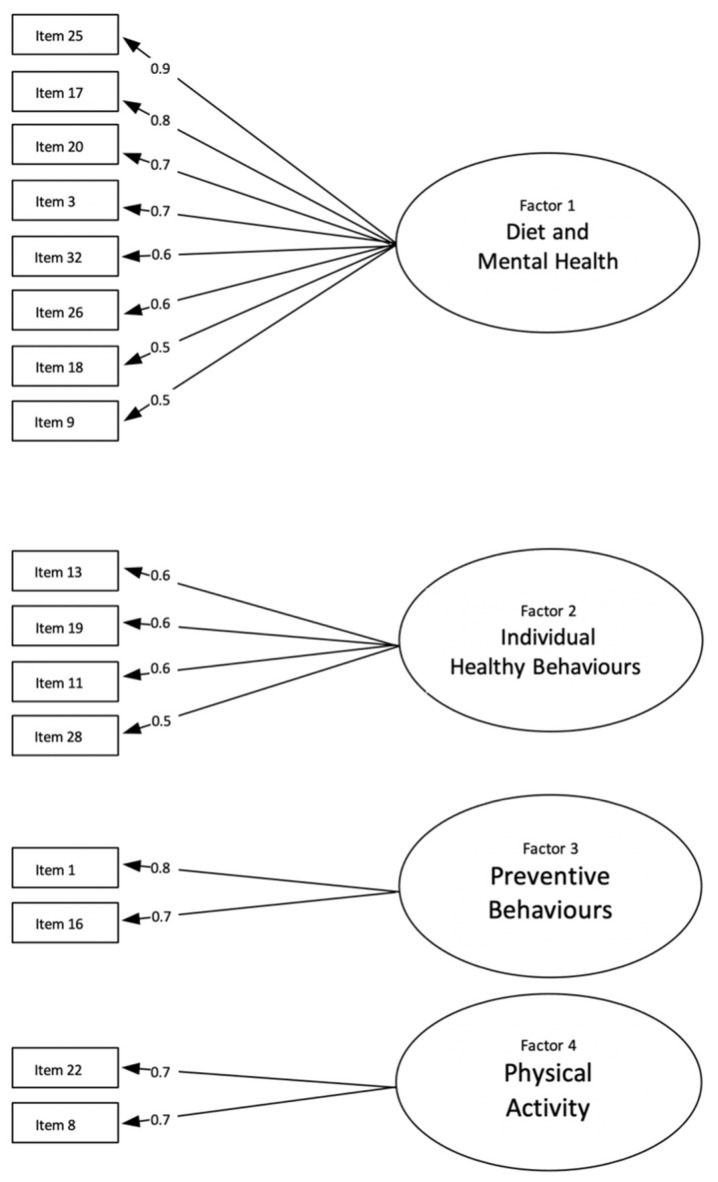
EFA of the HBS items.

**Table 1 behavsci-12-00378-t001:** Factor Loadings and Item Communalities by EFA (n = 487).

	Item	F1	F2	F3	F4	h^2^	u^2^	com
1	I check my blood sugar levels			0.84		0.72	0.28	1.0
3	I eat regularly	0.68				0.64	0.36	1.2
8	I lead an active lifestyle				0.72	0.54	0.46	1.1
9	My diet is varied	0.49				0.47	0.53	1.8
11	When buying food products, I check their composition		0.61			0.44	0.56	1.1
13	I look for information on healthy eating		0.64			0.52	0.48	1.1
16	I check my cholesterol levels			0.67		0.55	0.45	1.1
17	I find the time to relax and rest	0.78				0.65	0.35	1.1
18	I limit the consumption of sugar and foods which contain it (sweets)	0.51				0.36	0.64	1.3
19	I check my body for physical lesions or abnormalities		0.63			0.35	0.65	1.2
20	I eat breakfast	0.71				0.73	0.27	1.3
22	I use daily activities as an opportunity for physical activity (e.g., I climb the stairs instead of using the elevator, park my car at a distance so that I can walk, I move around by bicycle)				0.72	0.54	0.46	1.0
25	I provide my body with enough sleep	0.85				0.67	0.33	1.0
26	I limit the consumption of salt and foods which contain it	0.57				0.35	0.65	1.4
28	I perform a breast self-examination		0.47			0.25	0.75	1.1
32	I can effectively manage stress	0.57				0.25	0.75	1.4
	Eigenvalues	5.249	2.845	1.272	1.067			
	% of variance	0.22	0.10	0.08	0.07			
	Cumulative variance	0.22	0.32	0.40	0.47			

Only factor loadings > 0.4 are presented.

**Table 2 behavsci-12-00378-t002:** Means, standard deviations, reliability and HBS-16 factor correlation matrix (*p* < 0.01).

	M	SD	Cronbach’s Alpha	McDonald’s Omega	Factor 1	Factor 2	Factor 3	Factor 4
Factor 1 (8 items)	16.0	5.3	0.87	0.87	1.00			
Factor 2 (4 items)	6.5	3.0	0.69	0.70	0.76	1.00		
Factor 3 (2 items)	3.8	1.6	0.77	0.77	−0.56	−0.06	1.00	
Factor 4 (2 items)	3.6	1.7	0.68	0.69	−0.19	−0.44	0.01	1.00

## Data Availability

The datasets generated and analysed during the current study are available from the corresponding author.

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
