# Peer review of "Development and Validation of a Health Behaviour Scale: Exploratory Factor Analysis on Data from a Multicentre Study in Female Primary Care Patients"

_behavsci, 2022, doi:10.3390/bs12100378_

Round 1

Reviewer 1 Report

The article is relevant and presents an innovative proposal for a scale on health behaviors for women aged 45 and over. I would like to know more information about the literature review carried out, especially about the levels of evidence.

I suggest better explaining the conceptual difference between items 2 and 12, despite presenting the statistical results, the definition was not clear.

Reviewer 2 Report

The present study analyzes the psychometric properties of of a Health Behaviour Scale. First, I congratulate the manuscript's authors for their excellent work and effort. It seems essential to me that this type of research is published—next, my recommendations for the authors.

Introduction

1. The introduction is very well written. For this section, I have no recommendations. 

Method

2. Research Design. I recommend that you include a section entitled "research design". Recent classifications indicate that the studies that analyze the psychometric properties of scales are called Instrumental Designs (Ato, López & Benavente, 2013; Montero & León, 2007). I would also include the ethical aspects of the study in this section. Currently the authors have placed them in the participants section.

-Ato, M., López, J. J., & Benavente, A. (2013). A system of classification of research designs in psychology. Annals of Psychology, 29 (3), 1038-1059.

-Montero, I., & León, O. G. (2007). A guide for naming research studies in psychology. International Journal of Clinical and Health Psychology, 7 (3), 847-862.

3. ParticipantsIn this section the relevant characteristics of the sample should be established: What type of sample will we use in the study? What are the sociodemographic characteristics of the sample? It would help if the sociodemographic characteristics were in a frequency distribution table. What method will be used to obtain the sample? How many participants? What are the inclusion and exclusion criteria? 

Results

4. Internal Consistency. There are dozens of papers in the psychometric literature that have shown that the most popular measure of internal consistency reliability, Cronbach's alpha coefficient, is seriously flawed. For this reason, I recommend that you add an additional reliability index; preferably the Omega (McDonald, 1999).

Suggested Literature:

-Dunn, T., Baguley, T., & Brunsden, V. (2013, in press). From alpha to omega: A practical solution to the pervasive problem of internal consistency estimation. British Journal of Psychology.

-McDonald, R. P. (1999). Test theory: A unified approach. Mahwah, NJ: Lawrence Erlbaum Associates.

5. Internal Consistency of Factor 2 Factor 4:  I have serious reservations with the reliability of this subscales. Devellis (2017) suggests the following values to interpret internal consistency: below .60, unacceptable; between .60 and .65, undesirable; between .65 and .70, minimally acceptable; between .70 and .80, respectable; between .80 and .90, very good; and much above .90, one should consider shortening the scale. According to these classifications, this scale is below acceptable levels. Based on best practices for adapting and validating instruments, the authors should review the scale and evaluate whether eliminating certain items would increase internal consistency.  If by eliminating items, the internal consistency does not increase, the subscale should be eliminated. 

Discussion

6. This section should be reviewed once the authors make the new analyzes.

Reviewer 3 Report

Abstract

Triangulation, comparison with another instrument and refinement were not performed in the validation¿

Present the results of validation tests

Introduction

Link to QOL according to WHO and current definitions

There is a lack of robustness in the proposal of the study, only 1 paragraph reporting the gap in the literature, it is necessary to investigate other tests and justify with greater effusion the need to create a new instrument

Methods

Triangulation, comparison with another instrument?

Insert in the figure the validation phase - instrument refinement with experts

Introduce the instrument with appendix

Present the form of score accounting, what is the score of each item¿ how to perform the calculation¿ has syntax for the calculation available¿ the classification of the instrument (if any) in terms of different status in relation to the score

results

The analysis of the comparison with another instrument is lacking

Discussion

Deserves greater robustness of analysis and process results, not just comparing with existing studies. Discuss (based on the literature) the result of the study.
